# Chondromodulin is necessary for cartilage callus distraction in mice

**Kiminori Yukata[1,2]\***, **Chisa Shukunami[3,4]**, **Yoshito Matsui[1]**, **Aki Takimoto[4]**,
**Tomohiro Goto[1]**, **Mitsuhiko Takahashi** **[1]**, **Atsushi Mihara**[2], **Tetsuya Seto[2]**,
**Takashi Sakai[2]**, **Yuji Hiraki[4]**, **Natsuo Yasui[1]**

**1** Department of Orthopaedics, Institute of Biomedical Sciences, Tokushima University Graduate School, Tokushima, Japan, **2** Department of Orthopedic Surgery, Yamaguchi University Graduate School of Medicine, Yamaguchi, Japan, **3** Department of Molecular Biology and Biochemistry, Graduate School of Biomedical & Health Sciences, Hiroshima University, Hiroshima, Japan, **4** Department of Cellular Differentiation, Institute for Frontier Life and Medical Sciences, Kyoto University, Kyoto, Japan

\* kyukata2004jp@yahoo.co.jp

## Abstract

Chondromodulin (Cnmd) is a glycoprotein known to stimulate chondrocyte growth. We examined in this study the expression and functional role of *Cnmd* during distraction osteogenesis that is modulated by mechanical forces. The right tibiae of the mice were separated by osteotomy and subjected to slow progressive distraction using an external fixator. *In situ* hybridization and immunohistochemical analyses of the lengthened segment revealed that *Cnmd* mRNA and its protein in wild-type mice were localized in the cartilage callus, which was initially generated in the lag phase and was lengthened gradually during the distraction phase. In *Cnmd* null (*Cnmd*$^{-/-}$) mice, less cartilage callus was observed, and the distraction gap was filled by fibrous tissues. Additionally, radiological and histological investigations demonstrated delayed bone consolidation and remodeling of the lengthened segment in *Cnmd*$^{-/-}$ mice. Eventually, *Cnmd* deficiency caused a one-week delay in the peak expression of *VEGF*, *MMP2*, and *MMP9* genes and the subsequent angiogenesis and osteoclastogenesis. We conclude that Cnmd is necessary for cartilage callus distraction.

## Introduction

Chondromodulin (Cnmd, formerly Chondromodulin-I, Chm-1) is a cartilage-specific glycoprotein that stimulates proteoglycan synthesis in cultured growth plate chondrocytes and the colony formation of chondrocytes cultured in agarose [1, 2]. Some *in vitro* studies suggested the functional role of Cnmd as a chondrocyte growth modulator and an angiogenesis inhibitor [1–4]. Our previous study demonstrated that *Cnmd* null (*Cnmd*$^{-/-}$) mutation lead to less cartilage callus formation during the fracture healing process [5], although *Cnmd*$^{-/-}$ mice did not show abnormalities in cartilage development or endochondral bone formation during embryogenesis or normal growth, and further did not affect natural articular cartilage development [4, 6]. Thus, Cnmd functions as a chondrocyte modulator in specific conditions, causing osteogenesis such as cartilage or bone injury, but not in normal cartilage development and growth.

**Data Availability Statement:** All relevant data are within the paper and its Supporting information files.

**Funding:** This study was supported by Grants-in-Aid for Scientific Research from the Ministry of

Education, Culture, Sports, Science and Technology of Japan (No. 18390418 to N.Y. and No. 22591683 to Y.M.). The funders had no role in study design, data collection and analysis, decision to publish, or preparation of the manuscript.

**Competing interests:** The authors have that no competing interests exist.

It has been well established that mechanical stimuli also control cartilage development and homeostasis. For example, muscular paralysis in the chick embryo resulted in joint cavity formation failure, and dynamic compressive loading activates the biosynthesis of various extracellular matrices in the articular cartilage [7–10]. Mechanical stress determines the bone fracture healing throughout the endochondral or intramembranous bone formation, as the loss of stability leads to more cartilage callus formation [11, 12]. Interestingly, *Cnmd* regulated ossification patterns; bone callus formation increased, but cartilage callus decreased in *Cnmd*-deficient tibial fractures [5]. These findings strongly suggest the relationship between mechanical stress and Cnmd in the process of cartilage callus formation during bone repair/regeneration.

We focused on distraction osteogenesis, which involves an osteotomy followed by a slow progressive distraction to lengthen congenitally or traumatically shortened extremities, to assess the impact of the mechanical stress on *Cnmd* [13, 14]. Previous experimental studies on animals demonstrated abundant elongated cartilage callus at the earlier time point of the distraction phase, resulting in endochondral or transchondroid bone formation [15, 16]. Mechanical stress-induced various growth factors, including bone morphogenetic proteins, insulin-like growth factor-1, and transforming growth factor beta that underlies cartilage callus distraction [17, 18]. Therefore, animal models of distraction osteogenesis are useful *in vivo* experimental systems for examining the response to mechanical stress during bone regeneration.

This study aimed to determine the *Cnmd* response to mechanical stress in cartilage callus distraction. A recently established mouse model was chosen for this study due to the availability of knockout mice [19]. Taking advantage of this mouse model of tibial lengthening using an external fixator, we performed *in situ* hybridization, immunohistochemistry, and real-time reverse transcription-polymerase chain reaction (RT-PCR) to examine the expression levels and localization of *Cnmd* on the effect of mechanical tension stress. Furthermore, we compared the histogenesis characteristics in *Cnmd*$^{-/-}$ mice with those of wild-type littermates.

## Materials and methods

### Experimental animals

Mice harboring a null targeted mutation of the *Cnmd* gene were generated and maintained as previously described [4]. *Cnmd*$^{-/-}$ mice were backcrossed over eight times to C57/BL6 mice. In this study, nine to ten-week-old male *Cnmd*$^{-/-}$ mice and WT littermates (weighing between 23 and 27 g) were used. All experimental procedures were performed according to the protocol approved by the Laboratory Animal Care and Use Committee of The University of Tokushima Graduate School.

### Mouse model of distraction osteogenesis in the tibia

Mice were anesthetized with isoflurane. Following the established mouse model of distraction osteogenesis [19–21], an external fixator, consisted of two incomplete circular aluminum rings (outer diameter 20 mm, inner diameter 10 mm, thickness 1 mm), six nuts and two stainless screws (diameter 2 mm, length 25 mm), was applied to the right tibia using two 27G hypodermic needles for each proximal and distal ring (Fig 1). The total weight of the device, including four needles, was 4.0 g. A transverse osteotomy with the scalpel blade No. 11 was performed at the diaphysis of the tibia between the two rings. The wound was closed with 5–0 nylon sutures after osteotomy. The fibula was not broken at the surgery, but it was naturally broken around the epiphysis during the distraction phase and was also lengthened. The mice were kept in cages after recovery from anesthesia, allowing free unrestricted weight bearing. Following surgery, the animals were allowed to heal for 1 week (latency phase). After the latency phase, the

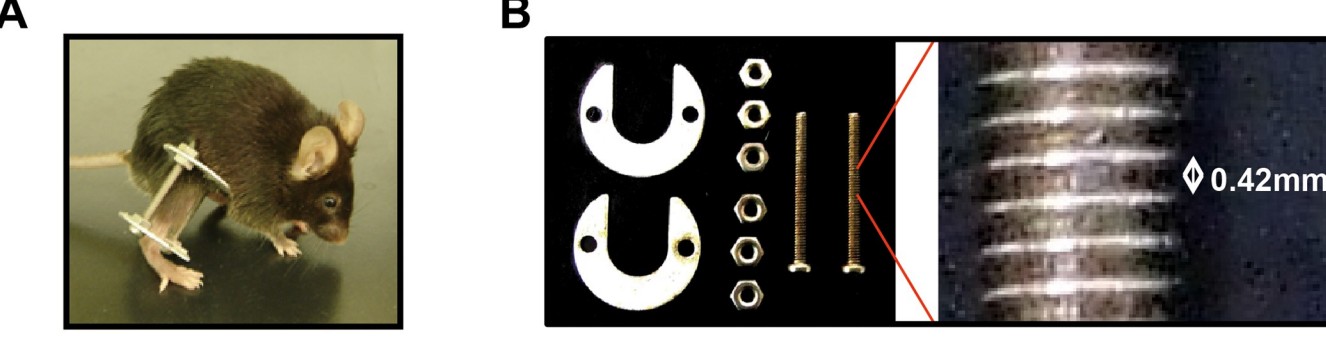

**Fig 1.** (A) The mouse model of distraction osteogenesis using an external fixator. (B) The external fixator consisted of two incomplete circular aluminum rings, six stainless steel nuts, and two stainless steel screws. The length between two pitches of screw was 0.42 mm. (C) The upper four nuts were turned half every twelve hours during the distraction phase to move the upper ring toward the proximal end of the screws by 0.21 mm. Finally, tibiae were lengthened by 5.88 mm after 2 weeks.

lengthening was initiated at a rate of 0.21 mm every 12 hours for 2 weeks (distraction phase). This was followed by a 9-week period of consolidation phase. Finally, tibiae were lengthened by 5.88 mm after 2 weeks. A total of 170 male mice (97 wild-type (WT) and 73 $Cnmd^{-/-}$ mice) were used. Radiographs were taken weekly with a soft radiograph apparatus (CMB-2, SOFTEX, Japan) under anesthesia (n = 10 for each WT and $Cnmd^{-/-}$). Radiographic assessment of bone union was performed using postoperative radiographs at 8 weeks after osteotomy. Bone union was defined as when two of two cortices were bridged, partial union was defined as when either side of the cortical bone was not bridged, and non-union was defined as when both two cortices were not bridged. Mice were euthanized with isoflurane and carbon dioxide. For histology, mice were sacrificed 1, 2, 3, 4, 6, 8, and 12 weeks after osteotomy (n = 4 for each genotype). For RNA extraction, mice were also sacrificed at 0, 1, 2, 3, 4, 5, 6, and 8 weeks after osteotomy (n = 3–5 for each genotype). One WT and two $Cnmd^{-/-}$ mice were euthanized postoperatively due to infection. One WT and one $Cnmd^{-/-}$ mice of them were scheduled to report histology 12 weeks after osteotomy, and one left *the Cnmd^{-/-}* mouse for RNA extraction 8 weeks after osteotomy. Finally, 167 mice were included in the present study.

## Tissue preparation and staining

The right tibiae were dissected with the surrounding muscles at the indicated times after osteotomy and fixed at 4˚C for 3 days with 4% paraformaldehyde in 0.1 M phosphate buffer (pH 7.4). They were decalcified with Morse solution (10% [wt/vol] sodium citrate and 22.5%

[vol/vol] formic acid) for *in situ* hybridization (n = 3 for only WT mice) or with 0.5 M ethylenediaminetetraacetic acid (pH 7.4) for immunohistochemistry and tartrate-resistant acid phosphatase (TRACP) staining (n = 3 for each genotype). The decalcified tissues were dehydrated through increasing concentrations of ethanol and embedded in paraffin. Serial sagittal sections, 6 μm thick for *in situ* hybridization or 4 μm thick for toluidine blue (pH 4.1), alcian blue hematoxylin & eosin-orange G staining, TRACP staining or immunohistochemistry, were mounted on MAS-coated slides (Matsunami Glass Industries, Japan). These sections were stored at 4°C until histological analyses. TRACP staining was performed using TRACP-staining kit (Hokudo, Japan), and counterstained with methyl green. The mean numbers TRACP-positive cells per 5 random areas (0.2 mm$^2$) in both proximal and distal remodeling zone were calculated manually in WT and *Cnmd*$^{-/-}$ mice using the image software Win ROOF ver 5.6.0 (Mitani Corp., Japan) and the mean numbers among 3 samples at each time point were analyzed.

## Cartilage callus area

The sagittal sections were stained with toluidine blue and areas showing metachromasia were identified as cartilage callus. The metachromatic cartilage callus area was measured manually in WT and *Cnmd*$^{-/-}$ mice using the image software Win ROOF ver 5.6.0 (Mitani Corp.) at 1, 2, and 3 weeks after osteotomy (n = 4 for each genotype).

## *In situ* hybridization

The antisense and sense RNA probes for each gene under analysis were transcribed from expression plasmids with a digoxygenin (DIG) RNA labeling kit (Roche Diagnostics, Mannheim, Germany). Hybridization was carried out at 50°C for 16 hours and the sections were washed under high stringency. The DIG-labeled molecules were detected using NBT/BCIP (Roche) as the substrate for anti-DIG antibody-coupled alkaline phosphatase. Sections were mounted with coverslips, and then photographed. For RNA probes, the cDNAs of rat *type II collagen alpha 1* (*Col2a1*), mouse *type X collagen alpha 1* (*Col10a1*) and a 1.0 kb mouse *Cnmd* were used as templates for *in vitro* transcription after linearization of the plasmids [5]. Three specimens were included for WT mice at each time point.

## Immunohistochemical staining

To examine the localization of Cnmd, type II collagen and type X collagen, and CD31 in the lengthened callus, immunohistochemistry was performed using an avidin-biotin peroxidase detection system (Vector Lab, Burlingame, CA, USA). After blocking in 5% skim milk in PBS-T (phosphate-buffered saline containing 1% Tween 20) at 4°C for 1 hour, the sections were incubated overnight at 4°C with a primary antibody, anti-Cnmd rabbit polyclonal antibody (1:1000), anti-type II collagen rabbit polyclonal antibody (1:400: Rockland Immunochemicals, Philadelphia, PA, USA), anti-type X collagen polyclonal antibody (1:400: LSL, Japan), and anti-CD31 rabbit polyclonal antibody (1:50: Abcam, Cambridge, MA, USA). After washing with PBS-T, the sections were incubated with secondary antibodies, a biotinylated anti-rabbit IgG (Vector Lab). Reactions were visualized with diaminobenzidine as substrate (Vector Lab). The sections for Cnmd, type II collagen, type X collagen, and CD31 were counterstained with hematoxylin. Five random fields (60000 μm$^2$) per lengthened segment were measured, on CD31 stained slides, to determine the number of newly formed vascular vessels for samples 3 and 4 weeks after osteotomy. Three specimens were included for each genotype at each time point.

### RNA isolation and real-time RT PCR analysis

The surrounding muscles and bone marrow were removed from the intact and lengthened tibiae at various postoperative time points for RNA extraction (n = 3–5 for each genotype). The samples were snap-frozen in liquid nitrogen and smashed using the TissueLyser II instrument (Qiagen, Hilden, Germany). Total RNA was isolated from each sample using Qiagen RNeasy Lipid Tissue Mini Kits according to the instructions provided by the manufacturer (Qiagen). Total RNA (1 μg) was then reverse transcribed to cDNA using the Superscript® III first strand synthesis system for the RT-PCR kit (Invitrogen). For quantitative analysis of the expression level of *Cnmd*, *Sex-determining region Y-box containing gene 9* (*Sox9*), *Col2a1*, *Col10a1*, *Vascular endothelial growth factor* (*VEGF*), *Matrix metalloproteinases* (*MMPs*)-2 and 9, 14, *Tenomodulin* (*Tnmd*), *Tissue inhibitor of metalloproteinase* (*TIMP*)-2 transcripts during distraction osteogenesis, real-time RT-PCR was performed in the ABI Prism 7500 (Applied Biosystems, Foster City, CA, USA) using pre-validated Assays on Demand™ (consisting of 20X mix of unlabelled PCR primers and Taq-Man® MGB probe (FAM™ dye labelled) (S1 Table). Levels of *Gapdh* transcripts were used to normalize each gene expression level.

### Statistical analysis

Data are shown as mean ± standard error (SE). Fisher's exact test was used for the radiographic assessment of union and partial union rates. The Mann-Whitney U test was used for the data from gene expression and histological analyzes. These tests were performed using the XLSTAT (Mindware Inc., Okayama, Japan). *P* values of <0.05 were considered significant.

## Results

### Delayed and incomplete fusion of the bone callus in *Cnmd*⁻/⁻ mice

Serial radiographs at various time points during distraction osteogenesis were obtained from WT and *Cnmd*⁻/⁻ mice (Fig 2). In WT mice, bone calluses appeared at the both ends of the lengthened area with a central radiolucent zone at 2 and 3 weeks after osteotomy. The bone callus developed from the periphery to the center of the lengthened area 4 weeks after osteotomy. Lengthened bone calluses were fused between 4 and 6 weeks after osteotomy, following remodeled until 12 weeks. In contrast, distraction osteogenesis in *Cnmd*⁻/⁻ mice demonstrated a less bone callus formation at 2 or 3 weeks after osteotomy, and their fusion and remodeling were delayed between 4 and 12 weeks.

Radiographic assessment of union, partial union, and nonunion rates demonstrated that 7 (70%) WT mice achieved bone union at 8 weeks after osteotomy compared with 1 (10%) *Cnmd*⁻/⁻ mice. Partial union was observed in 2 (20%) mice in WT mice, while in *Cnmd*⁻/⁻ mice partial union was observed in 8 (80%) mice. Non-union was observed in each 1(10%) mouse of both genotypes. The partial union rate was significantly higher in *Cnmd*⁻/⁻ mice than in WT mice (p<0.05).

### *Cnmd* and *Col2a1* are co-expressed in immature chondrocytes at the lengthened callus in WT mice

*In situ* hybridization was performed to determine whether the *Cnmd* mRNA was co-expressed with the *Col2a1* mRNA and the *Col10a1* mRNA, which are molecular markers of immature and hypertrophic chondrocytes, respectively. Histological examination of lengthened tibiae in WT mice showed development of a cartilage callus that was observed as toluidine blue-stained metachromasia near the osteotomy site at 1 week after osteotomy (Fig 3A). As the distraction progressed, the cartilage callus elongated along the tension vector. *Cnmd* mRNA was expressed in cells surrounded by the metachromatic cartilage matrix stained with toluidine blue. *The*

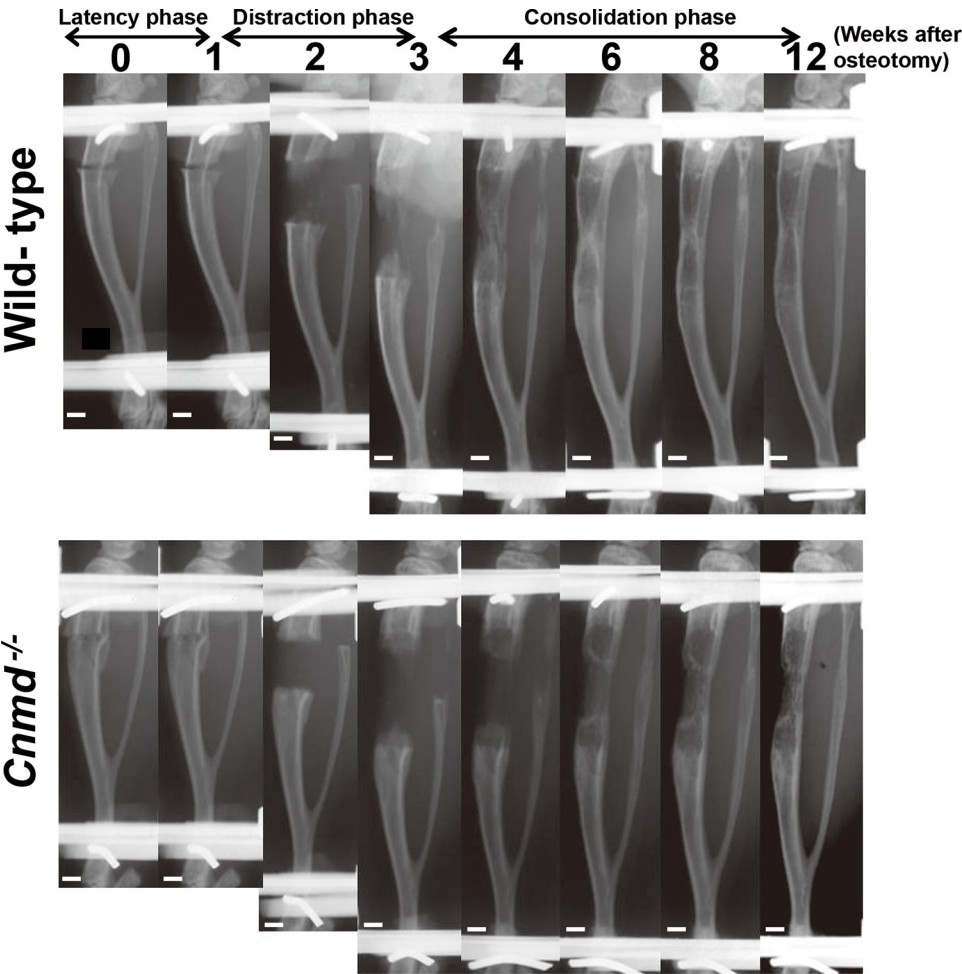

**Fig 2. Representative serial radiographs of osteotomized and lengthened tibiae in wild-type and *Cnmd*[-/-] mice at 0, 1, 2, 3, 4, 6, 8, and 12 after osteotomy.** Scale bar, 1mm.

*Cnmd* hybridization signals overlapped with the *Col2a1* expression areas, but not with the *Col10a1* expression area near the osteotomy site 1 week after osteotomy and in the lengthened segment 2 and 3 weeks after osteotomy. Real-time RT-PCR showed that *Cnmd* mRNA expression first appeared in the tibiae 1 week after osteotomy and had a maximum expression peak during the distraction phase (Fig 3B; $p < 0.05$ compared between 1 and 2 weeks). *Cnmd* expression levels declined between 3 and 8 weeks after osteotomy. The finding that *Cnmd* mRNA expression appears in the early stage of the distraction phase suggests a connection with chondrogenesis induced by mechanical stress. Immunohistochemistry demonstrated that the Cnmd protein was localized in the extracellular matrix of immature and hypertrophic chondrocytes, immunostained with anti-type II and X collagen antibodies, near the osteotomy site 1 week after osteotomy and in the lengthened segment 2 and 3 weeks after osteotomy (Fig 3C). At 4 weeks after osteoromy, the Cnmd protein was hardly detected.

## *Cnmd* null mutation resulted in fibrous callus formation during the early phases of distraction osteogenesis

We compared cartilage callus formation between WT and *Cnmd*[-/-] mice by histomorphometry and real-time RT-PCR because *Cnmd* existed in the cartilage callus. Cartilage callus formation

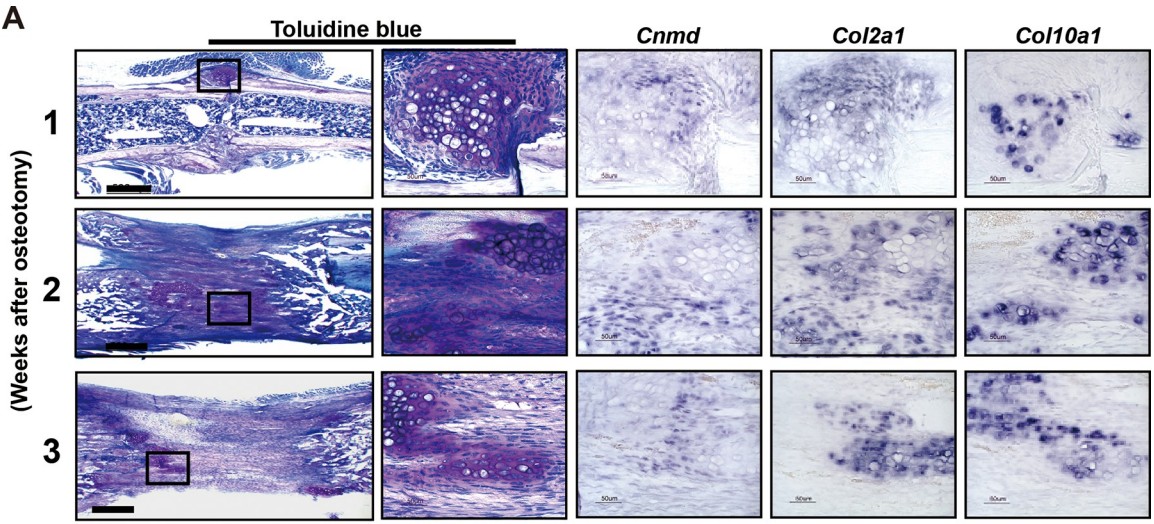

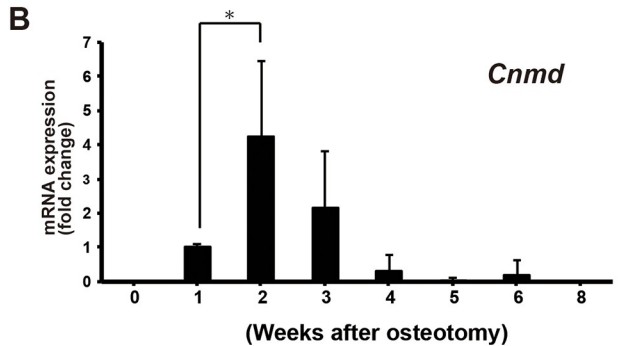

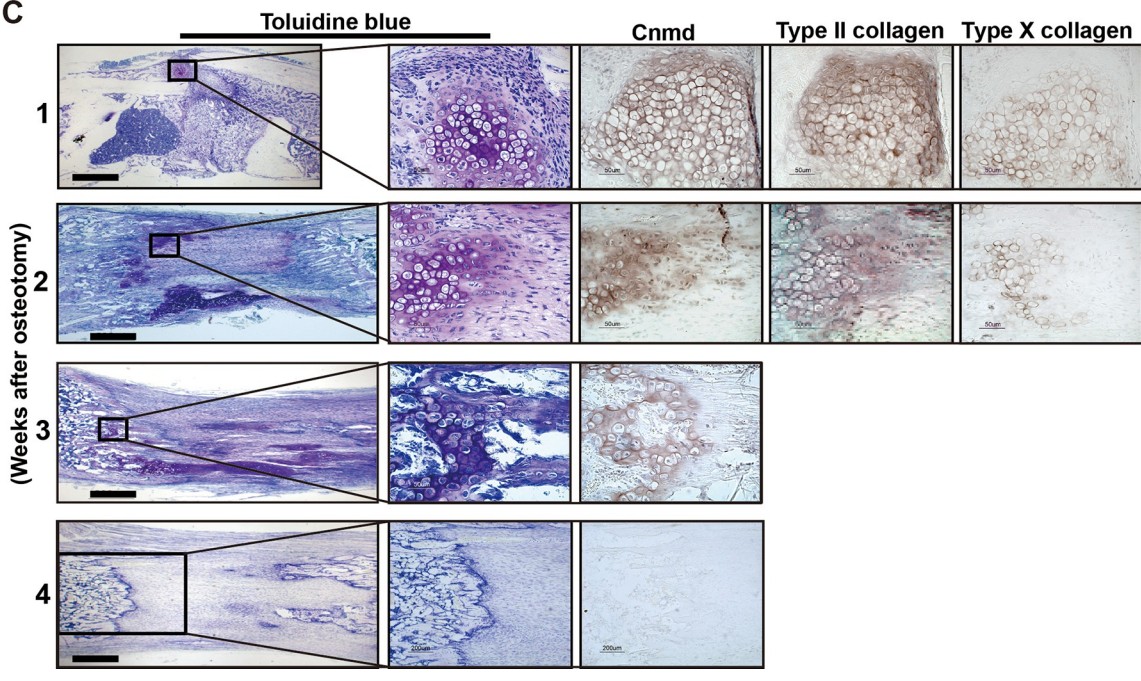

**Fig 3.** (A) mRNA localization of *Cnmd*. Sagittal sections stained with toluidine blue (left column) showing the representative osteotomized and lengthened tibiae at 1, 2 and 3 weeks after osteotomy with boxed areas of metachromatic cartilage callus. The adjacent sections were analyzed by *in situ* hybridization, demonstrating the expression of mRNA *Cnmd*, *Col2a1*, and *Col10a1* in the cartilage callus at 1, 2 and 3 weeks after osteotomy. Scale bar, 500 μm. (B) The time course change of *Cnmd* mRNA expression was measured by real-time RT-PCR, and was normalized to *Gapdh*. Data are shown as mean ± SE. *$p$ <0.05 between 1 and 2 weeks after osteotomy (C) Immunohistochemical location of *Cnmd*. Sections from the representative osteotomized and lengthened tibiae were stained with toluidine blue (left column) at 1, 2, 3 and 4 weeks after osteotomy, and areas of metachromatic cartilage callus were boxed. The adjacent sections were immunostained with anti-Cnmd, anti-type II collagen or anti-type X collagen antibodies. The brown signals in the right columns indicate the corresponding immunostainings. Scale bar, 500 μm.

in *Cnmd*$^{-/-}$ mice was reduced at 1 week after osteotomy, and some tissue defects were observed along with the fibrous callus formation in the lengthened segment at 2 weeks after osteotomy (Fig 4A). Interestingly, the tissue defects were fulfilled by the fibrous callus 3 weeks after osteotomy. Quantitative histomorphometry revealed that the cartilage area was significantly reduced in the callus of *Cnmd*$^{-/-}$ mice by 3.2-fold ($p$<0.05) at 1 week after osteotomy, 5.5-fold ($p$<0.05) at 2 weeks and 8.3-fold ($p$<0.05) at 3 weeks after osteotomy (Fig 4B). Real-time RT-PCR showed that *Col10a1* mRNA expression level was also significantly decreased in *Cnmd*$^{-/-}$ mice consistent with reduced toluidine blue-stained metachromasia cartilage callus formation, while *Sox9* and *Col2a1* mRNAs expression level was slightly increased at 1, 2, and 3 weeks after osteotomy in *Cnmd*$^{-/-}$ mice when compared to that of WT mice although there was no statistically significance (Fig 4C). These results indicate that *Cnmd* is indispensable for abundant cartilage callus formation. Besides, lack of *Cnmd* seems to alter the class of histogenesis during this period, from generation of cartilage callus to fibrous callus.

## Incomplete bone union and remodeling in *Cnmd*$^{-/-}$ mice

Histology revealed that bone calluses at both ends of the lengthened segment were united 4 to 6 weeks after osteotomy in WT mice, consistent with radiographic findings (Fig 5A). Then the united bone callus was gradually remodeled into mature bone from 6 to 12 weeks after osteotomy. On the contrary, partial bone union and delayed remodeling of the lengthened segment were observed until 12 weeks after osteotomy in *Cnmd*$^{-/-}$ mice.

## Delayed angiogenesis and lengthened tissue remodeling during distraction osteogenesis in *Cnmd*$^{-/-}$ mice

Angiogenesis and osteoclastogenesis are essential for bone healing and its remodeling processes. We compared newly formed vascular vessels in the lengthened segment and osteoclasts in the remodeling zone between WT and *Cnmd*$^{-/-}$ mice, and plotted the expression levels of angiogenic and anti-angiogenic genes in a time course. At 3 weeks after osteotomy, the capillary density in the lengthened segment in *Cnmd*$^{-/-}$ mice was significantly decreased when compared to that in WT mice (Fig 5B), while significantly increased at 4 weeks after osteotomy. Similarly, *the* null mutation of *Cnmd* was associated with a significant decrease in the number of TRACP positive cells at the end of the distraction phase, while the number of TRACP positive cells increased 4 weeks after osteotomy (Fig 5C). Real-time RT-PCR showed that *VEGF*, one of the key regulators of angiogenesis [23], decreased in *Cnmd*$^{-/-}$ mice 2 weeks after osteotomy, perhaps due to reduced endochondral bone formation. *VEGF* expression levels increased significantly 3 weeks after osteotomy, followed by enriched fibrous callus formation (Fig 5D). Expression of *MMP9*, an inducer of angiogenesis and tissue remodeling [22], reached a peak level at 2 weeks after osteotomy, when endochondral bone formation occurred in WT mice. Peak expression of *MMP9* did not occur until 3 weeks after osteotomy in *Cnmd*$^{-/-}$ mice. The expression pattern of these genes was consistent with the number of new vessels and

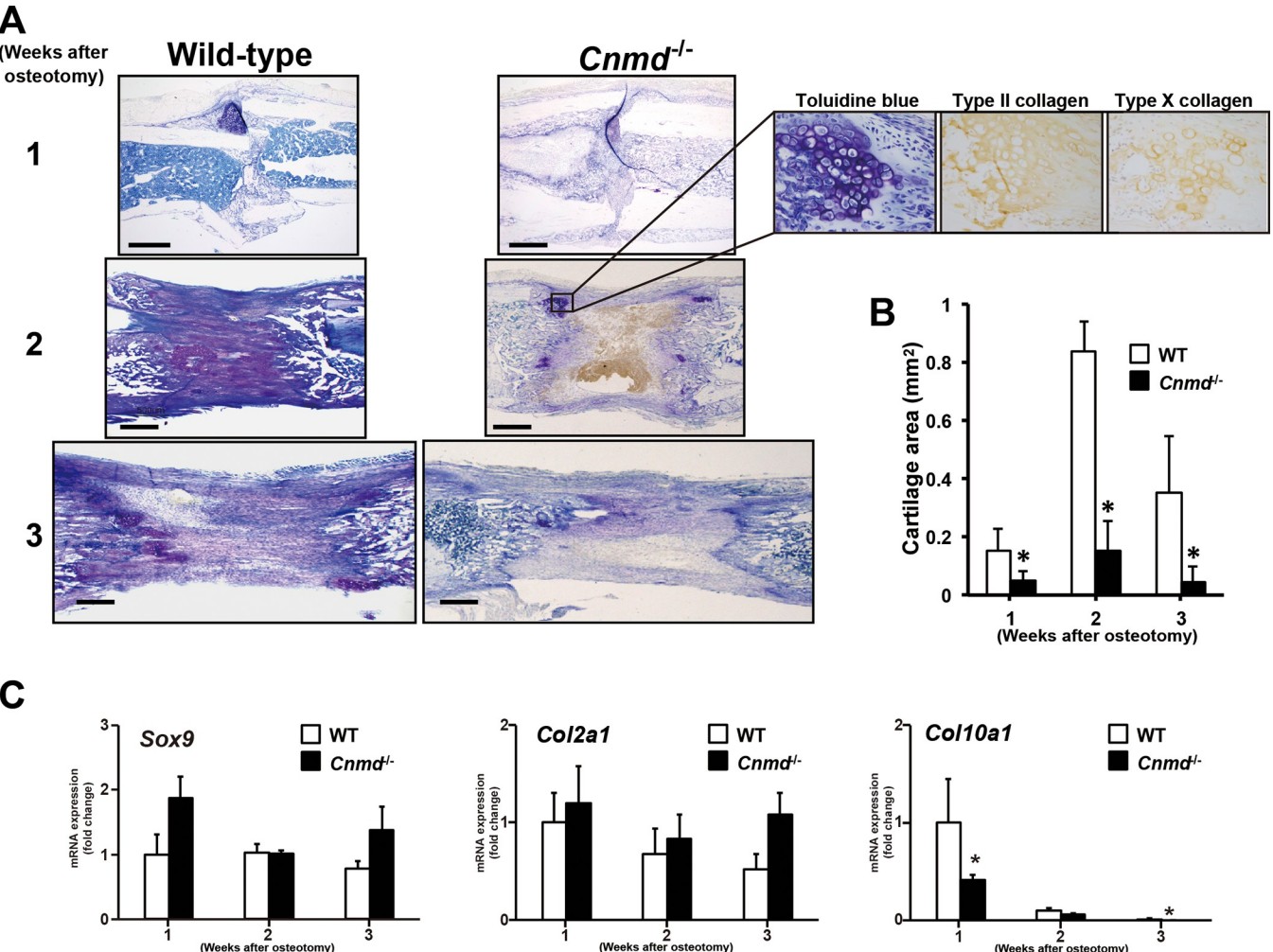

**Fig 4.** (A) Sagittal sections of representative osteotomized and lengthened tibiae of wild-type (WT) (left column) and *Cnmd⁻/⁻* (right column) mice at 1, 2, and 3 weeks after osteotomy, stained with toluidine blue. Areas showing metachromasia represent cartilage callus. Scale bar, 500 μm. (B) The callus areas of the cartilage in both genotypes at 1, 2, and 3 weeks after osteotomy were measured by histomorphometry. Data are shown as mean ± SE. (C) *Sox9*, *Col2a1* and *Col10a1* mRNA expression levels at 1, 2, and 3 weeks after osteotomy were measured by real-time RT-PCR, and were normalized to *Gapdh* was calculated. Data are shown as mean ± SE. Statistic examination was performed using ANOVA and significant differences are indicated by symbols: *$p < 0.05$, compared between WT and *Cnmd⁻/⁻* mice.

osteoclasts. Collectively, *Cnmd* deficiency caused a delay of 1 week in the peak expression of the *VEGF* and *MMP9* genes and subsequent angiogenesis and osteoclastogenesis. *MMP2* and *MMP14* also had delayed peak levels in *Cnmd⁻/⁻* mice. However, the expression of the antiangiogenic factors *Tnmd* and *TIMP2* reached a maximum level 3 weeks after osteotomy in WT and *Cnmd⁻/⁻* mice. *Tnmd* and *TIMP2* expression levels in *Cnmd⁻/⁻* mice were significantly higher between 2 and 4 weeks after osteotomy compared to WT.

## Discussion

Previous studies reported that Cnmd could directly stimulate chondrocyte proliferation and proteoglycan synthesis *in vitro* [1, 2]. Additionally, cyclic mechanical stimulation using cell stretching instruments improved chondrocyte proliferation and extracellular matrix mRNA expression [23–25]. These findings suggest that mechanical stress promotes chondrocyte

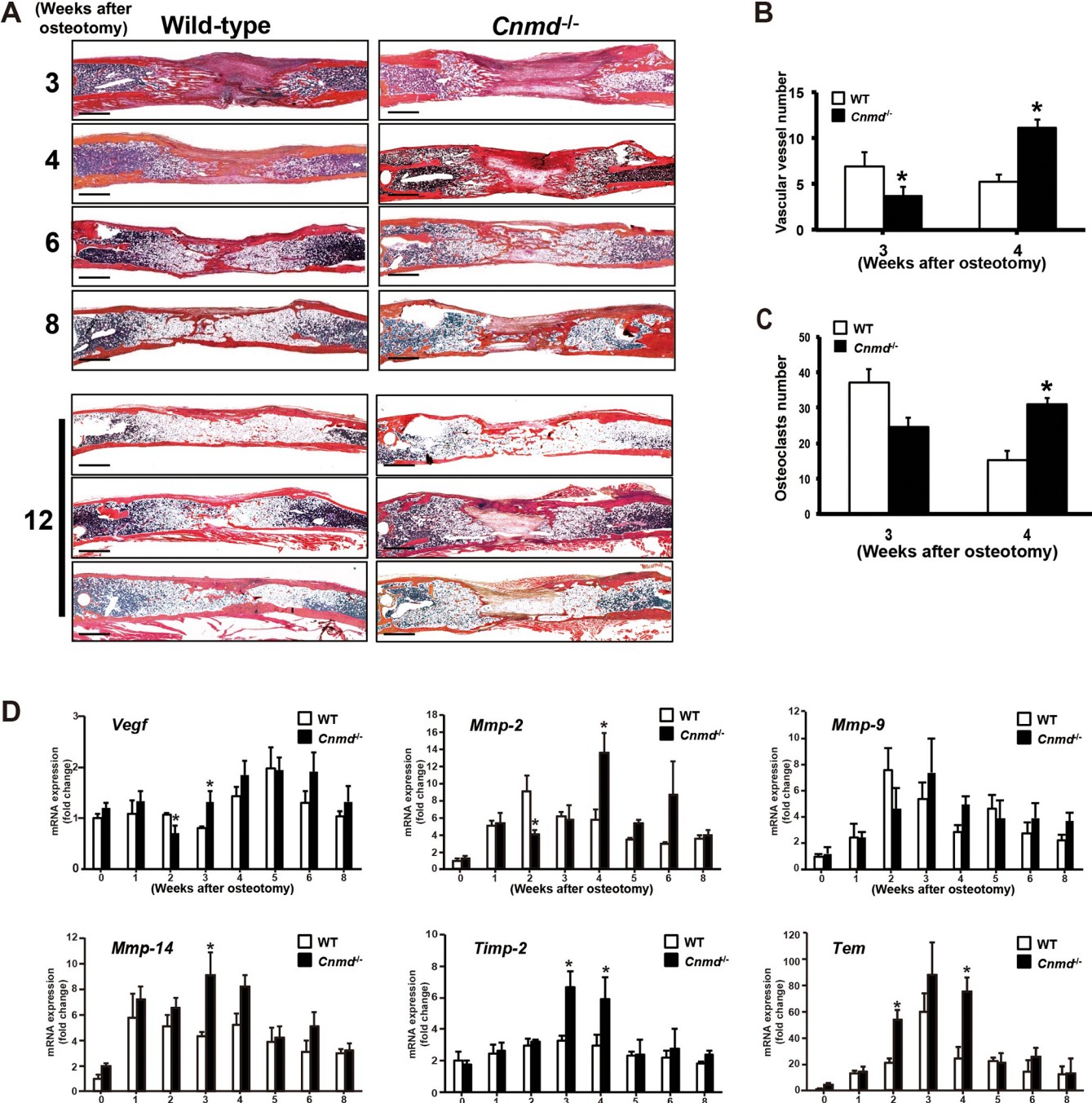

**Fig 5. Alcian blue and hematoxylin & eosin-orange G staining sections were prepared from wild-type (WT) and *Cnmd⁻/⁻* mice harvested at 3, 4, 6, 8, 12 weeks after osteotomy.** Representative lengthened tibiae were shown in each group. Three distinct lengthened tibiae were shown from each genotype at 12 weeks after osteotomy. Scale bar, 1 mm. (B) The capillary density (number of newly formed vascular vessels/field) in the lengthened segment at 3 and 4 weeks after osteotomy. (C) The number of osteoclasts (number of TRACP positive cells/field) in the remodeling zone at 3 and 4 weeks after osteotomy. (D) Time course change of anti-angiogenic and angiogenic gene expression in WT and *Cnmd⁻/⁻* mice. *VEGF*, *MMP2*, *MMP9*, *MMP14*, *Tnmd*, and *TIMP2* mRNA levels at 0, 1, 2, 3, 4, 5, 6 and 8 weeks after osteotomy were measured by real-time RT-PCR and normalized to *Gapdh*. Data are shown as mean ± SE. Statistic examination was performed using ANOVA and significant differences are indicated by symbols: $^*p < 0.05$, compared between WT and *Cnmd⁻/⁻* mice.

proliferation and cell matrix production by upregulation of the *Cnmd* gene. The current study indicated that *Cnmd* expression was enhanced by mechanical tension stress in immature chondrocytes within the cartilage callus. The *Cnmd* null mutation resulted in the suppression of cartilage callus formation in distraction osteogenesis, and the newly generated tissue was fibrous.

Tissues after failed articular cartilage repair procedures, such as abrasion arthroplasty, microfracture, autologous chondrocyte implantation, and periosteal transplantation, are usually composed primarily of fibrous connective tissue and fibrocartilage [26, 27]. The overlying repair tissue in treated with microfracture-treated articular cartilage defects was predominantly fibrocartilage and lacked *Cnmd* expression [28]. Klinger et al. stated that *Cnmd* stabilized the chondrocyte phenotype and inhibited the terminal differentiation at the articular cartilage repair site [29]. Chen et al. also showed a similar result in subcutaneously implanted cartilage graft tissues using *Cnmd* gene-modified mesenchymal stem cells [30]. Moreover, Zhu et al. reported that *Cnmd* was indispensable for the maintenance of cartilage homeostasis after subcutaneous implantation of cultured articular chondrocytes using *Cnmd* deficient mice [6]. By contrast, *Cnmd* deficiency did not interfered with cartilage regeneration in ectopic and articular cartilage defects. In this study, some type II and X collagen-positive cartilage callus formed during distraction osteogenesis in the *Cnmd* null mutation. Therefore, these data suggest that *Cnmd* is unessential for cartilage regeneration and callus formation, but is important for chondrocytes proliferation and inhibition of calcification (suppression of angiogenesis) in regenerated cartilage.

During distraction osteogenesis, cartilage callus formation is dictated by external fixator stiffness, tissue stiffness, and functional loading and is influenced by the mechanical environment within the lengthened segment. In WT mice, cartilage callus formed on the periosteal surface near the osteotomy during the lag phase and finite element analysis revealed hydrostatic tensile stress and immediate tensile strain applied to the same area [5, 31]. On the other hand, in $Cnmd^{-/-}$ mice, no cartilage callus was observed in the same area and cartilage callus was observed only in the osteotomy gap region between the bone ends where the high hydrostatic pressure was generated. These data suggest that *Cnmd* is required for cartilage callus formation due to tensile stress on the periosteum and is less involved in it due to hydrostatic pressure between the gaps. During the distraction phase, the lengthened segment is subjected to varying degrees of hydrostatic tensile stress, and the presence of *Cnmd* plays an important role in cartilage callus distraction. However, Waanders et al. showed that the gap tissue is subject to approximately 15% deformation (compression and tension) during walking in a rabbit tibial lengthening model, indicating that compression as well as tensile forces act on the gap tissue during the distraction phase [32]. Additionally, the fibula spontaneously fractured during the distraction phase and lengthened in the same manner as the tibia. This raises the concern that the presence of the fibula in the current tibial lengthening procedure greatly affected the mechanical environment of the lengthened segment of the tibia. Thus, the mechanical environment within the lengthened segment during the current tibial lengthening is complex, and further research is necessary for the relationship between *Cnmd* expression and various types of mechanical stress [33].

In this study, another significant finding was delayed union and remodeling of the newly generated tissue into mature bone during the consolidation phase in the absence of *Cnmd*. Despite the expectation that *Cnmd* deficiency promotes angiogenesis, the capillary density in the lengthened segment of $Cnmd^{-/-}$ mice decreased at the end of the distraction phase and the appearance of osteoclasts was delayed [3, 34]. Consistent with histological change, mRNA expression levels of angiogenic factors, *VEGF* and *MMP*s, decreased in the early stage of distraction and increased in the later stage of distraction. These findings might reflect the impact

of the fibrous callus initially generated due to the *Cnmd* null mutation, but not the cartilage callus. The real-time RT-PCR data showed the increased expression of *Tnmd* mRNA in *Cnmd*$^{-/-}$ mice at the distraction phase. *Tnmd* is a type II transmembrane protein that shares a cysteine-rich domain with *Cnmd* at the C-terminus [35]. A previous study demonstrated that the C-terminal region of *Tnmd* inhibited the VEGF-stimulated DNA synthesis and tube morphogenesis in Matrigel as much as *Cnmd* [36]. Dex et al. revealed that *Tnmd* mRNA expression was strongly upregulated by 5% axial cyclic strain in tendon stem/progenitor cells [37]. These findings suggest that the compensatory increase in *Tnmd* expression may predispose to fibrous callus formation. Furthermore, the increased mRNA expression level of another antiangiogenic factor, *TIMP2*, at the later stage of distraction in *Cnmd*$^{-/-}$ mice also might modulate the later vascular vessel formation in the lengthened segment [38]. Thus, the relationship between *Tnmd* and distraction osteogenesis is also a concern for further study.

This study has some limitations. First, microCT and biomechanical analyses were not performed, which would be more precise to evaluate bone union and remodeling than histology. Second, we did not show a direct *Cnmd* synthesis in response to mechanical tension stress in chondrocytes both *in vivo* and *in vitro*. Third, previous studies demonstrated an increase in intramembranous bone callus formation in *Cnmd*-deficient fractures and ectopic bone formation of the intervertebral disc in *Tnmd*$^{-/-}$*Cnmd*$^{-/-}$ mice [5, 39]. However, abundant bone formation in the bone marrow area did not appear in the current *Cnmd*$^{-/-}$ distraction osteogenesis. One possible explanation for this result is that callus distraction might pull out a newly formed bone callus. Finally, previous animal models have reported that intramembranous ossification is predominant as species size increases and the latency period shortens [40–42]. In this mouse model, elongated cartilage callus was abundantly induced, perhaps because the latency period was rather long (7 days). In other words, the results of this study suggest that *Cmnd* may have little effect under conditions in which bone lengthening are performed on predominantly intramembranous ossification.

In conclusion, we found that the elongation of cartilage callus during distraction osteogenesis was suppressed in *Cnmd*$^{-/-}$ mice and subsequent bone formation and remodeling slowed and partially failed. The study results indicate that *Cnmd*-mediated cartilage callus elongation is necessary for distraction osteogenesis and *Cnmd* could be a mechanical response chondrogenic factor. New insights into the function of *Cnmd* may establish this molecule as a candidate therapeutic agent for successful bone healing.

## Supporting information

**S1 Table. Pre-validated Assays on Demand™ (mix of unlabelled PCR primers and Taq-Man® MGB probe (FAM™ dye labelled)).**
(DOCX)

**S1 File.**
(ZIP)

## Acknowledgments

We are grateful to Ms. H. Sugiyama and Ms. C. Kuki for secretarial help.

## Author Contributions

**Conceptualization:** Kiminori Yukata, Chisa Shukunami, Yoshito Matsui, Natsuo Yasui.

**Data curation:** Tomohiro Goto, Atsushi Mihara, Tetsuya Seto.

**Formal analysis:** Atsushi Mihara.

**Investigation:** Kiminori Yukata, Tomohiro Goto, Mitsuhiko Takahashi.

**Methodology:** Aki Takimoto.

**Supervision:** Takashi Sakai, Yuji Hiraki, Natsuo Yasui.

**Visualization:** Kiminori Yukata.

**Writing – original draft:** Kiminori Yukata.

**Writing – review & editing:** Chisa Shukunami, Yuji Hiraki, Natsuo Yasui.

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
