## [Decision Letter · Decision Letter 0]

27 Jan 2022

PONE-D-21-40664Chondromodulin is Necessary for Cartilage Callus elongation and Complete Distraction Osteogenesis in MicePLOS ONE

Dear Dr. Yukata,

Thank you for submitting your manuscript to PLOS ONE. After careful consideration, we feel that it has merit but does not fully meet PLOS ONE’s publication criteria as it currently stands. Therefore, we invite you to submit a revised version of the manuscript that addresses the points raised during the review process.

We look forward to receiving your revised manuscript.

Kind regards,

Jose Manuel Garcia Aznar

Academic Editor

PLOS ONE

Journal Requirements:

[NO authors have competing interests.]

Reviewers' comments:

Reviewer's Responses to Questions

**Comments to the Author**

1. Is the manuscript technically sound, and do the data support the conclusions?

Reviewer #1: Partly

2. Has the statistical analysis been performed appropriately and rigorously? 

Reviewer #1: I Don't Know

3. Have the authors made all data underlying the findings in their manuscript fully available?

Reviewer #1: Yes

4. Is the manuscript presented in an intelligible fashion and written in standard English?

Reviewer #1: Yes

5. Review Comments to the Author

Reviewer #1: This paper aims to demonstrate that chondromodulin is necessary for cartilage callus elongation during distraction osteogenesis in mice. They applied the same distraction protocol to mice without chondromodulin (cnmd null group) and a control group. Radiographies, immunohistochemical analysis and other results reveals that a poor callus and less production of cartilage at the cnmd null group. Therefore, authors suggest that regulates chondromodulin mechanical stress induced cartilage callus formation that is necessary for successful bone lengthening.

The publication of this work could help to understand the impact of chondromodulin on chondrogenesis and the difference in cartilage properties with and without chondromodulin. However, the present manuscript needs major changes because the authors obviate that there is controversy about the type of ossification during osteogenic distraction. Studies can be found in which ossification is mainly endochondral, intramembranous or both. They should be cited. Many studies have reported that ossification during distraction osteogenesis is primarily intramembranous, especially in large animals and models closer to humans. This should be addressed in the introduction and discussion of the article, and taken into account as a limitation of the article.

Taking the above into account implies changing the focus of the manuscript on many points. For example, if there is no cartilage in the distraction callus, chondromodulin will not be necessary. In the title may be read “Chondromodulin is Necessary for … and Complete Distraction”. This would not be true in these cases, so it can confuse the reader. In my opinion, according to the outcomes, chondromodulin is being shown to be necessary to create cartilage distraction callus. However, I do not see it essential for carrying out the distraction, since in many cases the distraction is carried out by means of intramembranous ossification.

Another example, lines 271 to 273: “Cnmd null mutation resulted in the suppression of cartilage callus formation in distraction osteogenesis, and the newly generated tissue was fibrous”. This occurs in other studies with normal animals. Why? Discuss it in the article.

Other comments:

Lines 162 to 175, in the text, parenthesis structures are used within parentheses, which makes reading extremely difficult. This section must be organized in another way to be understandable. It is advisable to use tables, footnotes or other resources.

Lines 85 to 101 ¿How many animals were used for each sacrifice time point?

It is not clear how many time points and how many animals were used for each type of analysis. A table summarizing this information would be convenient at the beginning of the results section or end of M&M section.

6. PLOS authors have the option to publish the peer review history of their article (what does this mean?). If published, this will include your full peer review and any attached files.

Reviewer #1: No

---

## [Author Response · Author response to Decision Letter 0]

3 Apr 2022

Re: Manuscript ID: PONE-D-21-40664

Title: Chondromodulin is Necessary for Cartilage elongation and Complete Distraction Osteogenesis in Mice

We thank you and the reviewers for the very useful comments and your time for critical and careful review of our manuscript. 

We have responded in a point-by-point fashion to the comments from the Editor and Reviewers and indicated how the suggestions have been incorporated in the revised manuscript where appropriate. We have clearly highlighted all the changes in the manuscript for easy identification. We believe the manuscript has been significantly improved and hope that it is suitable for publication in PLOS ONE. 

We look forward to hearing from you at your earliest convenience. 

Yours Sincerely,

Kiminori Yukata, MD, PhD.

(On Behalf of All Co-authors)

Dear editors,

We modified our manuscript according to PLOS ONE's style requirement. And, we added some information followed by editor's request.

Responses to Reviewers’ Comments:

Reviewer #1: This paper aims to demonstrate that chondromodulin is necessary for cartilage callus elongation during distraction osteogenesis in mice. They applied the same distraction protocol to mice without chondromodulin (cnmd null group) and a control group. Radiographies, immunohistochemical analysis and other results reveals that a poor callus and less production of cartilage at the cnmd null group. Therefore, authors suggest that regulates chondromodulin mechanical stress induced cartilage callus formation that is necessary for successful bone lengthening.

The publication of this work could help to understand the impact of chondromodulin on chondrogenesis and the difference in cartilage properties with and without chondromodulin. However, the present manuscript needs major changes because the authors obviate that there is controversy about the type of ossification during osteogenic distraction. Studies can be found in which ossification is mainly endochondral, intramembranous or both. They should be cited. Many studies have reported that ossification during distraction osteogenesis is primarily intramembranous, especially in large animals and models closer to humans. This should be addressed in the introduction and discussion of the article, and taken into account as a limitation of the article.

Response: Thank you very much for your critical review.　As reviewer suggested, in some situations (e.g., larger animals or shorter latency periods), bone lengthening occurs by predominantly intramembranous bone formation. Cartilage callus elongation and endochondral ossification may not be important, and, chondromodulin may not be necessary in such cases.　About this, we have added some sentences as a limitation.

Please see Page 15, Line 307-311.

Taking the above into account implies changing the focus of the manuscript on many points. For example, if there is no cartilage in the distraction callus, chondromodulin will not be necessary. In the title may be read “Chondromodulin is Necessary for … and Complete Distraction”. This would not be true in these cases, so it can confuse the reader. In my opinion, according to the outcomes, chondromodulin is being shown to be necessary to create cartilage distraction callus. However, I do not see it essential for carrying out the distraction, since in many cases the distraction is carried out by means of intramembranous ossification.

Response: Thank you very much for your good suggestion.　 We changed the title to “Chondromodulin is necessary for cartilage callus elongation during distraction osteogenesis in mice”

Please see the new Title in the revised manuscript and cover letter.

Another example, lines 271 to 273: “Cnmd null mutation resulted in the suppression of cartilage callus formation in distraction osteogenesis, and the newly generated tissue was fibrous”. This occurs in other studies with normal animals. Why? Discuss it in the article.

Response: Thank you very much for your good question.　Tissues after failed articular cartilage repair procedure are usually composed of fibrous connective tissues and fibrocartilage, which could lack chondromodulin expression. So, we added some sentences about it in the discussion section. Please see Page 13, Line 265-269.

Other comments:

Lines 162 to 175, in the text, parenthesis structures are used within parentheses, which makes reading extremely difficult. This section must be organized in another way to be understandable. It is advisable to use tables, footnotes or other resources.

Response: Thank you very much for your suggestion.　 We added the supplemental Table 1 about PCR primers’ information.

Lines 85 to 101 ¿How many animals were used for each sacrifice time point?

It is not clear how many time points and how many animals were used for each type of analysis. A table summarizing this information would be convenient at the beginning of the results section or end of M&M section.

Response: Thank you very much for your suggestion.

We described the number of mice in detail, which were used in this study.

Please see Page 5 line 87-96.

“Radiographs were taken weekly with a soft radiograph apparatus (CMB-2, SOFTEX, Japan) under anesthesia (n=10 for each WT and Cnmd-/-). Radiographic assessment of bone union was performed using postoperative radiographs at 8 weeks after osteotomy. Bone union was defined as when two of two cortices were bridged, partial union was defined as when either side of the cortical bone was not bridged, and non-union was defined as when both two cortices were not bridged. Mice were euthanized with isoflurane and carbon dioxide. For histology, mice were sacrificed 1, 2, 3, 4, 6, 8, and 12 weeks after osteotomy (n (n=4 for each genotype). For RNA extraction, mice were also sacrificed at 0, 1, 2, 3, 4, 5, 6, and 8 weeks after osteotomy (n=4 for each genotype).”

---

## [Decision Letter · Decision Letter 1]

7 Jul 2022

PONE-D-21-40664R1Chondromodulin is necessary for cartilage callus elongation during distraction osteogenesis in micePLOS ONE

Dear Dr. Yukata,

Thank you for submitting your manuscript to PLOS ONE. After careful consideration, we feel that it has merit but does not fully meet PLOS ONE’s publication criteria as it currently stands. Therefore, we invite you to submit a revised version of the manuscript that addresses the points raised during the review process.It was really difficult to find reviewers for this work. So, please, try to address carefully all the points indicated by the reviewer.

We look forward to receiving your revised manuscript.

Kind regards,

Jose Manuel Garcia Aznar

Academic Editor

PLOS ONE

Reviewers' comments:

Reviewer's Responses to Questions

**Comments to the Author**

1. If the authors have adequately addressed your comments raised in a previous round of review and you feel that this manuscript is now acceptable for publication, you may indicate that here to bypass the “Comments to the Author” section, enter your conflict of interest statement in the “Confidential to Editor” section, and submit your "Accept" recommendation.

Reviewer #2: (No Response)

2. Is the manuscript technically sound, and do the data support the conclusions?

Reviewer #2: Partly

3. Has the statistical analysis been performed appropriately and rigorously? 

Reviewer #2: N/A

4. Have the authors made all data underlying the findings in their manuscript fully available?

Reviewer #2: Yes

5. Is the manuscript presented in an intelligible fashion and written in standard English?

Reviewer #2: Yes

6. Review Comments to the Author

Reviewer #2: Authors perform an in vivo analysis in a mice model to determine the role of chondromodulin in distraction osteogenesis. They do a careful study on the different tissues and their compositions in the distracted callus (histological, radiographic, inmunohistochemical studies). They found that chondromodulin inhibition reduce cartilage in the distracted callus.

The topic of the paper is interesting; authors recognise in the title that the conclusions are limited to the animal model they use (a mice model). In fact, there is much debate in literature about the possibility of a soft/cartilage callus under tensile load as in distraction osteogenesis. Authors state in the paper that the cartilage appears in small animals under this conditions, however they should discuss a little bit about the real mechanical environment in the mice tibia. In fact, I have some doubts about their external fixator capacity to isolate just tensile load in the callus while distracting, probably the compressive and bending loads are also important during the gait cycle. This aspect should be further discussed and the authors should try to determine the real mechanical environment in the callus.

Comments:

Lines 83-84 “The fibula was not broken” In figure 1 it seems that the fibula resulted broken due to the distraction itself, in time points 2-6 weeks after fracture. If this is not the case, the load sharing mechanism between tibia-fibula-fixator would be altered, please discuss a Little bit in the discussion section about the role of fibula.

Lines 85-87. The distraction protocol is described, please include also the total distracted length to make the manuscript easier to read (it is already included in figure 1 legend).

Line 39 Typo “osteogenesis”

Line 131 “in vitro” it should be in cursive fonts.

Figure 1. To use a nomenclature similar to the state of the art replace “Lag phase” by “Latency hase”

Figure 1. Include a scale bar in the radiograph

Lines 160, 243, 251… Acronym of Vascular Endothelial Growth Factor should be better rewritten with capital letters VEGF. Same comment for Matrix metallopeptidase (MMP).

Lines 309-313. One of the critical point of the study is cartilage formation in a distracted callus in contrast to previous studies in which intramembranous ossification occur under distraction “perhaps because the latency period was rather long”, the latency period is long for a mice model but other factors could also favour endochondral ossification for example the external fixator as described in the paper could produce some bending and compression movement during the gait of the anima. Please, comment about it also.

7. PLOS authors have the option to publish the peer review history of their article (what does this mean?). If published, this will include your full peer review and any attached files.

Reviewer #2: No

---

## [Author Response · Author response to Decision Letter 1]

26 Aug 2022

Responses to Reviewers’ Comments:

Reviewer #2: Authors perform an in vivo analysis in a mice model to determine the role of chondromodulin in distraction osteogenesis. They do a careful study on the different tissues and their compositions in the distracted callus (histological, radiographic, inmunohistochemical studies). They found that chondromodulin inhibition reduce cartilage in the distracted callus.

The topic of the paper is interesting; authors recognise in the title that the conclusions are limited to the animal model they use (a mice model). In fact, there is much debate in literature about the possibility of a soft/cartilage callus under tensile load as in distraction osteogenesis. Authors state in the paper that the cartilage appears in small animals under this conditions, however they should discuss a little bit about the real mechanical environment in the mice tibia. In fact, I have some doubts about their external fixator capacity to isolate just tensile load in the callus while distracting, probably the compressive and bending loads are also important during the gait cycle. This aspect should be further discussed and the authors should try to determine the real mechanical environment in the callus.

Response: Thank you very much for your comment. 

In a rabbit tibial lengthening model, Waanders et al. showed that the gap tissue was exposed not only to tension force but also to cyclic strain of about 15% of the tension force during walking [40].

It means that not only tensile load but also various mechanical forces affect the distraction segment.

So, we added some sentences in the discussion section (Page 15, Line 315-320).

Comments:

Lines 83-84 “The fibula was not broken” In figure 1 it seems that the fibula resulted broken due to the distraction itself, in time points 2-6 weeks after fracture. If this is not the case, the load sharing mechanism between tibia-fibula-fixator would be altered, please discuss a Little bit in the discussion section about the role of fibula.

Response: Thank you very much for your critical comment. As the reviewer pointed out, we did not break the fibula, but it was naturally broken and lengthened during the distraction period. (Page5, Line 83-84) So, we believe that the mechanical influence of fibula on tibial lengthening is small. 

Lines 85-87. The distraction protocol is described, please include also the total distracted length to make the manuscript easier to read (it is already included in figure 1 legend).

Response: We added the total distracted length in the materials and method section. (Page 5, Lines 87-88)

Line 39 Typo “osteogenesis”

Response: We exchanged the word ‘chondrogeneses’ to ‘osteogenesis.’ (Page 3, Line 39)

Line 131 “in vitro” it should be in cursive fonts.

Response: Thank you very much. We corrected it. (Page 7, Line 132)

Figure 1. To use a nomenclature similar to the state of the art replace “Lag phase” by “Latency hase”

Response: Thank you very much for your suggestion. We replaced by ‘Latency phase’ throughout the manuscript and figures. (Line 85, 86, and Figure 2)

Figure 1. Include a scale bar in the radiograph

Response: We added a scale bar in the radiographs of figure 2. Also, we described about it in the figure legends.

Lines 160, 243, 251… Acronym of Vascular Endothelial Growth Factor should be better rewritten with capital letters VEGF. Same comment for Matrix metallopeptidase (MMP).

Response: Thank you very much for your critical review. We changed it as the reviewer suggested. Please see the corrected words marked in yellow.

Lines 309-313. One of the critical point of the study is cartilage formation in a distracted callus in contrast to previous studies in which intramembranous ossification occur under distraction “perhaps because the latency period was rather long”, the latency period is long for a mice model but other factors could also favour endochondral ossification for example the external fixator as described in the paper could produce some bending and compression movement during the gait of the anima. Please, comment about it also.

Response: Thank you very much for your comment. 

In a rabbit tibial lengthening model, Waanders et al. showed that the gap tissue was exposed not only to tension force but also to cyclic strain of about 15% of the tension force during walking [40].

So, we believe that some bending and compression movement in addition to tension stress would affect cartilage callus formation.

---

## [Decision Letter · Decision Letter 2]

10 Oct 2022

PONE-D-21-40664R2Chondromodulin is necessary for cartilage callus elongation during distraction osteogenesis in micePLOS ONE

Dear Dr. Yukata,

Thank you for submitting your manuscript to PLOS ONE. After careful consideration, we feel that it has merit but does not fully meet PLOS ONE’s publication criteria as it currently stands. Therefore, we invite you to submit a revised version of the manuscript that addresses the points raised during the review process.

Above all, please, try to improve the discussion of your results, it is a clear aspect to be improved.

We look forward to receiving your revised manuscript.

Kind regards,

Jose Manuel Garcia Aznar

Academic Editor

PLOS ONE

Reviewers' comments:

Reviewer's Responses to Questions

**Comments to the Author**

1. If the authors have adequately addressed your comments raised in a previous round of review and you feel that this manuscript is now acceptable for publication, you may indicate that here to bypass the “Comments to the Author” section, enter your conflict of interest statement in the “Confidential to Editor” section, and submit your "Accept" recommendation.

Reviewer #2: (No Response)

2. Is the manuscript technically sound, and do the data support the conclusions?

Reviewer #2: Yes

3. Has the statistical analysis been performed appropriately and rigorously? 

Reviewer #2: Yes

4. Have the authors made all data underlying the findings in their manuscript fully available?

Reviewer #2: No

5. Is the manuscript presented in an intelligible fashion and written in standard English?

Reviewer #2: Yes

6. Review Comments to the Author

Reviewer #2: In general, I find authors have answer to my previous questions but they should justify their answers and more important they should include some discussion in the paper. Also, please highlight the modified parts in the manuscript.

In my previous review I commented “Lines 83-84 “The fibula was not broken” In figure 1 it seems that the fibula resulted broken due to the distraction itself, in time points 2-6 weeks after fracture. If this is not the case, the load sharing mechanism between tibia-fibula-fixator would be altered, please discuss a Little bit in the discussion section about the role of fibula.” And authors answered “Response: Thank you very much for your critical comment. As the reviewer pointed out, we did not break the fibula, but it was naturally broken and lengthened during the distraction period. (Page5, Line 83-84) So, we believe that the mechanical influence of fibula on tibial lengthening is small”

They recognize in the paper that the fibula was broken during distraction, but they do not include any discussion about this in the discussion section, I think the mechanical environment in the fracture site is modified due to the broken fibula the authors just answer to me “we believe that the mechanical influence of fibula on tibial lengthening is small”, please justify and include some comments in the discussion section.

I also commented about the possible bending and compression movement in the gap, even though they recognize it should exit, nevertheless there is no discussion in the paper. They say

Lines 315-317 “Waanders et al. showed that the gap tissue was exposed not only to tension force but also to cyclic strain of about 15% of the tension force during walking” Please rephrase, what do you mean by cyclic strain (tension or compression), one should assume the tension force will result in tension strain. Please rephrase.

Typo:, line 106 “in situ” cursive.

7. PLOS authors have the option to publish the peer review history of their article (what does this mean?). If published, this will include your full peer review and any attached files.

Reviewer #2: No

---

## [Author Response · Author response to Decision Letter 2]

19 Nov 2022

Title: Chondromodulin is necessary for cartilage callus distraction in mice

We thank you and the reviewers for the very useful comments and your time for critical and careful review of our manuscript. 

We have responded in a point-by-point fashion to the comments from the Editor and Reviewers and indicated how the suggestions have been incorporated in the revised manuscript where appropriate. We have clearly highlighted all the changes in the manuscript for easy identification. We believe the manuscript has been significantly improved and hope that it is suitable for publication in PLOS ONE. 

We look forward to hearing from you at your earliest convenience. 

Yours Sincerely,

Kiminori Yukata, MD, PhD.

(On Behalf of All Co-authors)

Responses to Reviewers’ Comments:

Reviewer #2: In general, I find authors have answer to my previous questions but they should justify their answers and more important they should include some discussion in the paper. Also, please highlight the modified parts in the manuscript.

In my previous review I commented “Lines 83-84 “The fibula was not broken” In figure 1 it seems that the fibula resulted broken due to the distraction itself, in time points 2-6 weeks after fracture. If this is not the case, the load sharing mechanism between tibia-fibula-fixator would be altered, please discuss a Little bit in the discussion section about the role of fibula.” And authors answered “Response: Thank you very much for your critical comment. As the reviewer pointed out, we did not break the fibula, but it was naturally broken and lengthened during the distraction period. (Page5, Line 83-84) So, we believe that the mechanical influence of fibula on tibial lengthening is small”

They recognize in the paper that the fibula was broken during distraction, but they do not include any discussion about this in the discussion section, I think the mechanical environment in the fracture site is modified due to the broken fibula the authors just answer to me “we believe that the mechanical influence of fibula on tibial lengthening is small”, please justify and include some comments in the discussion section.

I also commented about the possible bending and compression movement in the gap, even though they recognize it should exit, nevertheless there is no discussion in the paper. They say

Lines 315-317 “Waanders et al. showed that the gap tissue was exposed not only to tension force but also to cyclic strain of about 15% of the tension force during walking” Please rephrase, what do you mean by cyclic strain (tension or compression), one should assume the tension force will result in tension strain. Please rephrase.

Response: Thank you very much for your careful review and pointing out.

The reviewer recommended us to discuss about the mechanical environment of the lengthened segment. That’s a great idea. So, we added one paragraph about the discussion on it from line 281 to 300. Actually, the extent to which the fibula mechanically affected the tibial lengthening is unclear, but it definitely affected. So we mentioned it in the discussion section.

Typo:, line 106 “in situ” cursive.

Response: Thank you very much for your suggestion. We modified it in Line 106.

Finally, we changed the title of this manuscript to ‘Chondromodulin is necessary for cartilage callus distraction in mice.’

Also, we changed some words, which were highlighted yellow.

Because ‘callus distraction’ is more common than ‘callus elongation’

---

## [Decision Letter · Decision Letter 3]

5 Jan 2023

Chondromodulin is necessary for cartilage callus distraction in mice

PONE-D-21-40664R3

Dear Dr. Yukata,

We’re pleased to inform you that your manuscript has been judged scientifically suitable for publication and will be formally accepted for publication once it meets all outstanding technical requirements.

Kind regards,

Jose Manuel Garcia Aznar

Academic Editor

PLOS ONE

Additional Editor Comments (optional):

Reviewers' comments:

Reviewer's Responses to Questions

**Comments to the Author**

1. If the authors have adequately addressed your comments raised in a previous round of review and you feel that this manuscript is now acceptable for publication, you may indicate that here to bypass the “Comments to the Author” section, enter your conflict of interest statement in the “Confidential to Editor” section, and submit your "Accept" recommendation.

Reviewer #2: All comments have been addressed

2. Is the manuscript technically sound, and do the data support the conclusions?

Reviewer #2: Yes

3. Has the statistical analysis been performed appropriately and rigorously? 

Reviewer #2: Yes

4. Have the authors made all data underlying the findings in their manuscript fully available?

Reviewer #2: Yes

5. Is the manuscript presented in an intelligible fashion and written in standard English?

Reviewer #2: Yes

6. Review Comments to the Author

Reviewer #2: The authors have addressed all my concerns. In my opinion the paper is now ready for publication in PLOS One.

7. PLOS authors have the option to publish the peer review history of their article (what does this mean?). If published, this will include your full peer review and any attached files.

Reviewer #2: No

---

## [Editor Report · Acceptance letter]

7 Feb 2023

PONE-D-21-40664R3 

Chondromodulin is necessary for cartilage callus distraction in mice 

Dear Dr. Yukata:

I'm pleased to inform you that your manuscript has been deemed suitable for publication in PLOS ONE. Congratulations! Your manuscript is now with our production department. 

Kind regards, 

on behalf of

Dr. Jose Manuel Garcia Aznar 

Academic Editor

PLOS ONE